# Effect of St. John’s Wort (*Hypericum perforatum* L.) on Male Sexual and Reproductive Health: A Narrative Review

**DOI:** 10.3390/biomedicines11102800

**Published:** 2023-10-16

**Authors:** Meshari A. Alzahrani, Salman Bin Ofisan, Nasser I. Alshumaymiri, Muath Alghuwainem, Muath Altamimi, Ali Y. Alali, Muhammad Rabie, Ahmed K. AboSkena, Khalid Almaymuni, Raed Almannie, Saleh Binsaleh

**Affiliations:** 1Department of Urology, College of Medicine, Majmaah University, Al-Majmaah 11952, Saudi Arabia; 2College of Medicine, Prince Sattam Bin Abdulaziz University, Al-Kharj 16273, Saudi Arabia; 3College of Science, Alexandria University, Alexandria 21568, Egypt; 4Department of Pharmaceutical, College of Pharmacy, Ahram Canadian University (ACU), 6th of October 12451, Egypt; 5College of Medicine, Majmaah University, Al-Majmaah 11952, Saudi Arabia; 6Division of Urology, Department of Surgery, Faculty of Medicine, King Saud University Medical City, King Saud University, Riyadh 11461, Saudi Arabia

**Keywords:** St. John’s wort, sexual health, reproductive health, premature ejaculation, male

## Abstract

Background: Hypericum species are widely acknowledged for their biological attributes, with notable attention being paid to *Hypericum perforatum*, commonly known as St. John’s wort (SJW) within the Hypericum section of the Hypericaceae family. This species is among the most thoroughly investigated herbal medicines, particularly in terms of its application in the management of mild to moderate depression. SJW is used to treat depression, menopausal symptoms, attention-deficit hyperactivity disorder (ADHD), somatic symptom disorder, obsessive–compulsive disorder, and skin conditions, such as wounds and muscle pain. However, the usefulness and effectiveness of SJW for male sexual and reproductive health (SRH) are not well known. Objective: To assess the current evidence in the literature on the effect of SJW on male SRH. Methods: This narrative review followed a predetermined protocol and used MEDLINE and PubMed to identify articles published in English on the effects of SJW on male SRH. The search used various keywords, such as “*Hypericum Perforatum*”, “St. John’s Wort”, and terms related to sexual and reproductive health issues. Articles published between the inception of the database and August 2023 were included. Results: We identified 12 articles published from 1999 to 2019, the majority of which were experimental and conducted on animals. These studies demonstrate variability in terms of design, sample size, type of SJW extract used, the dosage administered, and duration of treatment. Studies have indicated potential sexual dysfunction (SD) due to SJW, which includes reduced libido, delayed ejaculation, delayed orgasm, and erectile dysfunction. Additionally, reproductive toxicity has been suggested, as evidenced by spermicidal effects through the inhibition of sperm motility, abnormal spermatozoa, chromosomal aberrations, and DNA denaturation. Furthermore, some studies have reported potential adverse events during maternal exposure, inhibition of fertilization, and disruption of reproductive parameters. Conclusions: Our review suggests that the safety and efficacy of SJW in the treatment of human SRH remain unclear. Further comprehensive, well-designed studies with larger samples, longer exposure periods, and specific dosages are needed to clarify SJW’s effects of SJW. Therefore, consultation with healthcare professionals before using herbal remedies or supplements is crucial.

## 1. Introduction

*Hypericum perforatum* L., generally known as St. John’s wort (SJW), is a herbaceous perennial plant that was once regarded for its putative therapeutic properties, including wound-healing and diuretic, antimicrobial, and antiviral effects [1]. SJW is a plant that is native to Asia and Europe [2]. The name hypericum is said to have been acquired from the Greek words hyper and eikon, which mean over and image, and the SJW name may have been given because the flowers bloom near St. John’s Day, which is on the 24th of June [2]. The extracts and/or components of SJW have been demonstrated to possess a variety of pharmacological activities, including antibacterial, antidepressant, and antiviral effects [2]. SJW is also claimed to have anesthetic and astringent effects and has historically been used to treat conditions such as excitability, neuralgia, sciatica, menopausal neurosis, anxiety, and depression, as well as in topical formulations for treating wounds [3]. SJW contains several chemicals, including phloroglucinol derivates (hyperforin, adhyperforin), flavonoids (rutin, hyperoside, isoquercitrin, quercitrin), biflavonoids (biapigenin, amentoflavone), and naphthodianthrones (hypericin, pseudohypericin) [4,5]. The most widely used SJW forms of antidepressants are hydroalcoholic extracts of the aerial portion of the plant, which contain several natural compounds, including flavonoids, proanthocyanidins, naphthodianthrones (e.g., hypericin and pseudohypericin), and acylphloroglucinols (e.g., hyperforin and adhyperforin) [1,6]. SJW has numerous pharmacological effects, including the downregulation of beta receptors, increase in 5-HT2 receptors, weak inhibitory effects on monoamine oxidases (MAO) A and B, and significant inhibitory effects on the synaptosomal uptake of serotonin, noradrenaline, dopamine, gamma-aminobutyric acid (GABA), and L-glutamate [7]. Hyperforin (HF), an essential component of antidepressants, suppresses the uptake of 5-hydroxytryptamine (5-HT), dopamine (DA), norepinephrine (NA), gamma-aminobutyric acid (GABA), and L-glutamate into synaptosomal preparations [8,9]. However, hypericin, which has only been extracted from a few plants, has attracted much attention because it was initially described as a monoamine oxidase (MAO) inhibitor [10]. It was initially thought that the MAO-inhibitory properties of hypericin were responsible for the antidepressant effects of SJW extracts [10]. However, the concentrations of the extracts used to achieve this effect were too high (>100 µg/mL) to be achieved in vivo, and no effect on MAO activity was detected ex vivo in rats given 100 mg/kg of the whole extract [11,12]. Furthermore, the inhibition of monoamine reuptake by HF involves changes in intracellular H+ and Na+ concentrations and/or neurotransmitter storage in synaptic vesicles [13]. It was later found that SJW extracts inhibited the synaptosomal uptake of 5-HT, DA, and NA in vitro with high potency [14]. These findings imply that indirect action on sigma receptors may be responsible for the antidepressant effect of SJW extract [15]. Based on the available data, SJW is well tolerated, and its side effects are often mild; gastrointestinal symptoms, dizziness, confusion, and fatigue or sedation are the most frequently described adverse effects [2]. Furthermore, SWJ appears to be a safer choice than standard antidepressants for short-term use [2]. Because of the antidepressant-like effects of SJW extract, HF has been shown in animal studies to have an inhibitory effect on ejaculatory function [16]. Therefore, we believe that SJW has a strong effect on ejaculation, which is an essential component of male sexual behavior typically accompanied by orgasmic sensations [17]. When appropriate stimuli are received, a reflex is triggered in the spinal ejaculation generator (SEG), which coordinates the prostate, seminal vesicles, urethra, and pelvic floor muscles to expel semen from the body [18]. The neural circuit responsible for ejaculation relies on neurotransmitters and neuropeptides such as dopamine, glutamic acid, nitric oxide, and oxytocin. In contrast, serotonin and opioid peptides decrease activity in this circuit [19]. Several reports have shown that the activation of D2 receptors in the supraspinal ejaculation generator (SSEG) increases SEG excitability, while augmentation of 5-HT at these sites reduces SEG excitability [20,21]. The inhibitory effect of 5-HT on the neural circuit of ejaculation may explain the significant impact of chronic use of tricyclic antidepressants (TCAs) and selective serotonin reuptake inhibitors (SSRIs) [22]. Due to the potential side effects of lowered libido, SD, and blood pressure changes, the chronic use of SSRIs is not considered an ideal treatment option for PE. SJW is considered a natural product that is known to influence serotonin reuptake while producing fewer side effects and could be used as an alternative approach [4]. An increasing number of inventories containing natural or herbal substances are being explored in initial preclinical and clinical trials, for their potential to hinder ejaculation function is expanding. However, phytotherapeutic approaches such as SJW will form the foundation for approved remedies intended to address conditions such as premature ejaculation [23]. Evaluating sexual function, particularly in individuals with underlying comorbidities, can present a significant challenge as it becomes increasingly challenging to discern the causal relationship. Antidepressants, especially SSRIs, can often lead to SD, affecting desire, arousal, and performance. Discussing these side effects with a healthcare provider is crucial for exploring alternative treatments to manage depression and sexual issues. Hence, this review aims to explore the current state of knowledge regarding the therapeutic efficacy and safety of SJW on male sexual and reproductive health (SRH).

## 2. Materials and Methods

This review was conducted using a predefined protocol. Medline/PubMed was used to identify articles related to SJW’s effect of SJW on male SRH published in English. The following keywords were used to search through articles: “*Hypericum Perforatum*”, AND “*Hypericum perforatum* L.”, AND “St. John’s Wort”, AND “St. John’s Wort Dry Extract”, AND “premature ejaculation”, AND “ejaculation disorder”, AND “delayed ejaculation”, AND “orgasm disorder”, AND “delayed orgasm”, AND “sexual enhancement”, AND “sexual dysfunction”, AND “erectile dysfunction”, AND “male infertility”, AND “semen”, AND “sperm”. We searched for articles published between the date of the inception of the database and August 2023. Articles that were not in English, non-relevant articles, duplicate articles, abstract-only articles, no full-text articles, and books were excluded. Two senior authors (M.A.A. and R.M.) independently collected relevant information, including author names, publication year, study design, sample size, extract of SJW, dosage, and period of SJW use, and the main outcome reported on the effect of SJW on SRH. In cases of disagreement, a third reviewer was asked to resolve the matter by consensus. As this study did not involve human participants, there was no need for institutional review board approval or informed consent. Figure 1 shows a flowchart for selected studies that are included in this review.

## 3. Results

In our review, we identified 12 articles published between 1999 and 2019 focusing on the effects of SJW on male SRH. To ensure comprehensive coverage, we looked into these publications’ reference lists and most recent reviews in addition to reading the articles themselves. These studies differed in terms of design, sample size, SJW extract type, dose, and duration of treatment. The reviewed studies primarily yielded outcomes in Table 1. However, the results vary and are conflicting. Some studies have suggested that SJW may lead to sexual dysfunction, such as reduced libido, delayed ejaculation, orgasm, or erectile dysfunction. Moreover, some studies have indicated that reproductive toxicity manifests as a spermicidal effect through inhibition of sperm motility, abnormal spermatozoa, chromosomal aberrations, and DNA denaturation. Additionally, some studies have reported potential adverse events during maternal exposure, inhibition of fertilization, and disruption of reproductive parameters. Short-term SJW administration seemed to have minimal impact on circulating androgens, but it might reduce specific 5α-reduced androgens. Other studies have found no significant differences in maternal or infant demographics or adverse maternal events. Furthermore, some studies suggested that H. perforatum exhibited strong antioxidant properties and acted as a potent Deoxyribonuclease Inhibitor (DNase I), which could have prevented male infertility (Table 1).

Most of these studies were experimental and conducted on animals, with only a limited number of human subjects. The reviewed articles consistently exhibited small sample sizes, short treatment durations, and variations in SJW dosages.

## 4. Discussion

Antidepressant medicines such as TCAs and SSRIs are known to cause SD. According to recent research, SD develops in 30–60% of patients treated with antidepressants [35], and antidepressants with potent serotonergic characteristics have the highest risk of adverse sexual effects [36]. Antidepressant medications have been associated with decreased libido, impaired ejaculation, and inhibition of orgasm, ED, and priapism, with orgasm and ejaculation being more impaired than erection [35,36]. Some of these adverse effects, particularly those involving sperm transport and emission, may be caused by changes in the contractility of the smooth muscle of the vas deferens. For example, TCAs and SSRIs reduce human vas deferens motility [36,37]. Several surveys have revealed that a significant portion of the population (approximately a quarter of the population) in the United States (US) and Europe uses herbs to cure medical illnesses [38]. SJW, an effective antidepressant, is one such herb. SJW is by far the most frequently prescribed antidepressant in Germany, with physicians prescribing it four times more frequently than fluoxetine [39,40]. In the US, where herbs are typically accessible over the counter, SJW is one of the top seven herbal supplements, with retail sales (excluding grocery shops, drugstores, and mass market retail sales) estimated to be more than USD 14 million in 2002 [40]. Randomized controlled trials (RCTs) have shown SJW’s efficacy of SJW in the treatment of mild to moderate depression [39,40,41]. SJW has been shown to be therapeutically similar to imipramine or fluoxetine but with fewer adverse effects [39]. In contrast, case report findings have revealed that SJW may cause SD, including ED and orgasmic latency [26,27]. SJW contains various biologically active components, including naphthodianthrone hypericin, phloroglucinol derivatives such as HF, and flavonoids. The main active element in SJW is HF, which inhibits the reuptake of monoamines, such as serotonin, noradrenaline, and dopamine, as well as the amino acid neurotransmitters gamma-aminobutyric acid (GABA) and glutamate [38,39,40]. As SJW is commonly used to treat depression and has been linked to orgasmic delay [26], we evaluated the effect of this herbal treatment on male SRH. Furthermore, the mechanism of action of SJW and the effects of its key active ingredients are explored in this review. Our review shares the available English literature on the effects of SJW on male SRH over the last two decades. Our review is the first of its kind in the English literature to address the details of the effects of SJW and its treatment properties on male SRH, including its efficacy under the following conditions: PE, SF, and male fertility.

### 4.1. Impact of SJW on Male Sexual Health

Our review mainly focuses on the effect of SJW on PE, its direct effect on PE, and its underlying mechanism. The effects of SJW on male SF and female reproduction (Table 1).

One study examined the effectiveness of an HF-rich extract of *H. perforatum* against the pro-ejaculatory effects of 8-hydroxy-2-(diN-propylamino) tetralin (8-OH-DPAT) in a rat model of ejaculation, which mimics the expulsion phase of the ejaculatory reflex under anesthesia [16]. At lower doses, the HF extract inhibited the effects of 8-OH-DPAT on ejaculation, implying that HF acts either at the spinal ejaculation generator or directly on the neurons innervating the rhythmic bulbospongiosus (BS) muscles, which implies that HF can delay the ejaculatory reflex [16]. 

During the experiment, vigorous expulsion of urethral contents occurred simultaneously with the visual and mechanical characteristics of ejaculation. Engorgement and flaring of the penis glans were observed, and forceful squirts of liquid were ejected during ejaculation, distinct from the dripping that occurred during urine voiding [16]. A gradual reduction in both the amplitude and number of discharges was observed during the exhaustion of the ejaculatory reflex. In rats treated with vehicle (*n* = 6 in the control group), the administration of 8-OH-DPAT resulted in a significant acceleration of the ejaculation reflex. 8-OH-DPAT administration led to a significant increase in both the burst amplitude and duration of electrical bursts from baseline. The increase in burst amplitude was 203.2% ± 32.9%, while the increase in duration was 178.1% ± 22.9% (n = 6 in the treated group) [16]. However, the effects of 8-OH-DPAT on ejaculation were considerably diminished when the subjects were pretreated with the HF extract. The efficacy of the HF extract was higher at lower doses in mg/kg (range, 5–80 mg/kg). The most effective dose (10 mg/kg) was selected for further experiments. HF extract significantly reduced the increase in the amplitude of electric discharges induced by 8-OH-DPAT from the baseline by 13.4% ± 4.1%. Furthermore, the duration of bursts in the BS muscles also showed a decrease of 37.6% ± 9.6% due to the use of HF extract [16]. This compound has the ability to non-competitively inhibit the reuptake of serotonin, noradrenaline, dopamine, gamma-aminobutyric acid, and glutamate [4,42].

TCAs and SSRIs are prescribed off-label for the management of PE, in addition to their primary use as antidepressant drugs [43]. The side effects commonly experienced by patients taking TCAs and SSRIs, such as changes in libido, impaired ejaculation, and inhibited orgasms, may lead some patients to seek natural or herbal therapies for PE. This may be particularly desirable for men with PE. Although HF has been found to have the potential to delay the ejaculatory reflex, further studies are necessary to fully understand its effects [16]. Nevertheless, because *H. perforatum* has a history of safe use as a nutritional supplement, the HF-enriched extract could be considered a treatment option for men with PE. However, two cases showed that SJW may cause SD, including erectile dysfunction, orgasmic latency, and delayed ejaculation [26,27]. SJW is suitable for specific patients with depression symptoms, and it is essential to undergo a psychiatric evaluation before considering its inclusion. However, in a case reported by Assalian et al., the patient had a known case of chronic recurrent depression on SSRIs sertraline daily dose of 100 mg. On sertraline, the patient complained of orgasmic latency, suppressed sexual desire, and ED, and the most annoying issue was the latter. He also experienced delayed ejaculation with masturbation, and sertraline was stopped, and SJW 2 tablets (0.9 mg each) were started twice daily. In this case, the main cause of his SD is using the sertraline because once the sertraline was discontinued, a few days later, the patient restored his SF, and the patient continued to take SJW, but one week later, he started developing ED, and they offered to start PDE5i (sildenafil 50 mg) was started along with SJW and instructed to take sildenafil one hour before the sexual intercourse, resulting in a good erection and effective intercourse. Assalian et al. also stated that they were aware of additional patients using SJW, but no SD were documented [26]. Another case report by Bhopal et al. reported a patient with a history of anxiety, depression, and obsessive–compulsive disorder (OCD); he was treated with different antidepressants, including fluoxetine, buspirone, bupropion, moclobemide, and paroxetine, but have been discontinued due to side effects, then he started on SJW and his depressive symptoms considerably abated, after 9 months he starts developing low libido and more depressive symptoms, following that, the SJW was replaced with 20 mg of citalopram daily. His sexual libido reverted [26,27].

Assalian et al. case report study reported a dose of SJW that was not previously reported, but the case report of Bhopal et al. did not. When comparing SJW to SSRIs, it has been shown that SJW has a less therapeutical effect on major depression, according to a clinical trial study [44]. In these cases, we assume that the nature of the psychiatric illness and the chronic use of antidepressants may play a role in the severity of symptoms, including provoked SD. In addition, it is difficult to determine whether SJW was the actual cause of SD in the mentioned case reports. Owing to the nature of the study design and the very small sample size, the results were considered inconclusive. 

According to experimental research, the primary active component of SJW is HF, which blocks the reuptake of monoamines, including serotonin, noradrenaline, and dopamine, as well as amino acid neurotransmitters γ-aminobutyric acid and glutamate [38,39,40]. Clinical trials utilizing extracts of SJW with various HF concentrations show that the anti-depressive action of SJW is dependent on its HF level [45]. It is interesting to note that HF decreases the sodium gradient by triggering a sodium conductivity mechanism that has not yet been characterized, which then inhibits the uptake of neurotransmitters [13]. Instead of being a selective inhibitor of either the synaptic membrane or vesicular monoamine transporters, HF interferes with the storage of monoamines in synaptic vesicles [46]. Another hypothesis is that HF may interfere with the uptake of neurotransmitters by eradicating the pH gradient created by the outflow of protons pushed inward across the synaptic vesicle membrane [47]. HF was the main component of the SJW extract that inhibited absorption, as confirmed by the failure of all other examined constituents to demonstrate substantial dependence [48] due to the fact that SJW is frequently used to treat depression and since it has been shown to induce an orgasmic delay [26]. There have been in vitro studies on this herbal remedy’s impact on the contractility of the rat and human vas deferens. In addition, research has been conducted on SJW’s potential mode of action of SJW as well as the impact of its primary active ingredients. SJW appears to offer major advantages over traditional antidepressants in that it is associated with fewer side effects and has been demonstrated to reduce the symptoms of mild-to-moderate depression [39,40,41].

Two case reports have drawn attention to the risk that the SJW plant could cause SD [26,27] or when it is related to delayed ejaculation with masturbation [26]. The nonselective profile of inhibition suggests that SJW directly affects the contractility of smooth muscle (post-junctional effect) rather than the mechanisms behind the release of neurotransmitters [30]. It was discovered that SJW and HF both prevented the human vas deferens from contracting in response to phenylephrine [30]. After therapeutic SJW dosing, it is probable that changes in vas deferens function could take place in vivo [30]. In vitro research demonstrated for the first time that the antidepressant SJW directly suppresses the contractility of rat and human smooth muscle in the vas deferens [30]. The delay in sperm emission observed in individuals receiving SJW may be partially explained if validated in vivo [30].

It has been shown that SJW protects human neuroblastoma cells from H2O2-induced apoptosis and reduces free radical production in both cell-free and human vascular tissue [49]. This may promote the efficacy of SJW in males at the erectile tissue level. A pilot study reported the effects of SJW extract on plasma androgen concentrations in healthy men and women [31]. Short-term SJW treatment has no effect on the concentration of most circulating androgens in men and women, although it may cause a reduction in some circulating 5-alpha-reduced androgens [31]. Healthy volunteers (six males and six females) were used in this study. They were observed before and after a 14-day treatment period with an SJW preparation that has previously been shown to stimulate CYP3A4 activity. Plasma concentrations of testosterone, dihydrotestosterone (DHT), dehydroepiandrosterone sulfate (DHEAS), sex hormone-binding globulin (SHBG), androsterone sulfate (AoS), and epiandrosterone sulfate (epiAoS) combined plasma concentrations. The combined concentrations of the 5-reduced steroids, AoS, and epiAoS significantly decreased after treatment in all participants (*p* = 0.02) and in males (*p* = 0.04). The results showed that SJW did not significantly change the majority of the androgens tested (*p* > 0.05). Furthermore, both men and women had higher testosterone-to-dihydrotestosterone ratios than men did. Even while the latter rise fell short of statistical significance, it is nonetheless consistent with SJW’s potential to inhibit 5α-reductase. They concluded that short-term treatment of SJW does not significantly change the concentrations of the majority of circulating androgens in men and women despite strongly inducing CYP3A4, although it may cause a diminution in certain of the circulating 5α-reduced androgens [31]. It should be noted that this pilot study had a small sample size of only 12 human subjects involved) six males and six females and a relatively short SJW dosing period of 14 days, limiting the power to detect minor changes in plasma androgen concentrations.

### 4.2. Impact of SJW on Male Reproductive Health

Few studies have investigated the effects of SWJ on sperm motility, and the results have been mixed and conflicting, both in vivo and in vitro.

SJW causes mutations in sperm cells [24]. It is also known to severely injure the kidneys and liver of Wistar rats, and its levels increase during the breastfeeding period. Liver and kidney lesions were more severe when the dose was increased from 100 to 1000 mg/kg per day and when the animals were breastfed for 21 days. The findings of this study indicate that additional histological studies in different animal species are needed to better assess the safety of Hypericum extract administration during pregnancy and breastfeeding [29]. Concerns have been raised about the use of SJW during pregnancy; indeed, pregnant and lactating women regularly take the SJW and other herbal therapies because they are typically popularly considered “natural and hence safe” [50]. However, there are no systematic data on the effects of Hypericum during pregnancy or breastfeeding. In contrast, a study comparing 30 breastfeeding women taking different SJW preparations for postnatal depression with other breastfeeding mothers who were not taking SJW found several adverse infant events, including two cases of colic, two cases of drowsiness, and one case of lethargy in the SJW group [28]. Other studies found no notable negative effects in mothers or infants from sporadic hypericum use during pregnancy (two patients) or lactation (one patient) [51,52]. In 2013, an animal study found that adult male rats administered 100 mg/kg SJW daily during pregnancy and lactation showed no effect on reproductive parameters [33]. Long-term studies on pregnancy and antenatal and breastfeeding outcomes are required before prescribing SJW for the treatment of depression in women. Despite the broad availability and use of SJW, as well as significant available research, there is little data on its reproductive safety. Currently, SJW is not recommended as a safe therapy during pregnancy or breastfeeding.

The observation study of Ondrizek et al. in 1999 showed SJW has a spermicidal impact and mutagenic on human sperm cells, as evidenced by suppression of sperm motility and SWJ decreased sperm motility at high concentrations (0.6 mg/mL) [24]. They concluded that SJW was the most potent inhibitor of sperm motility. Because sperm require motility for transit, penetration of cumulus cell layers, and engagement with oocytes, several elements of fertility may be jeopardized, especially in the presence of high levels of SJW. Further research is needed to establish the exact mechanism of action of this herb on sperm cells [24]. A limitation of this experimental study is that it used only sperm from a single fertile donor for a short period of time, reaching up to 48 h. Therefore, it is difficult to draw a clear conclusion on whether SJW affects human sperm parameters. Additional research is required to elucidate the exact mode of action of SJW on sperm cells. In 1999, the same author investigated zona-free hamster oocytes incubated for one hour in a control medium with saw palmetto (*Serenoa repens*), *Echinacea purpura*, *Ginkgo biloba*, or SWJ before sperm-oocyte contact [25]. Pretreatment of oocytes with 0.6 mg/mL led to a penetration. No differences were observed at the lower concentration (0.06 mg/mL). Reduced oocyte penetration was also observed at high echinacea and ginkgo concentrations. DNA denaturation is an outcome of sperm exposure to *Echinacea purpura* and SWJ. In contrast, saw palmetto and ginkgo had no effect. The sperm that was exposed to 0.6 mg/mL SJW had a BRCA1 exon 11 gene mutation. Oocytes were negatively affected by high doses of SWJ, echinacea, and ginkgo. No changes were made to the saw palmettos. SWJ, ginkgo, and echinacea can harm reproductive cells when administered in large doses. Sperm cells are mutagenic to SWJ [25]. A limitation of this study is that it used human sperm from a single fertile donor as well as one hamster with approximately 12–15 oocytes per group. Although medical practitioners frequently depict the procedure as incapable of generating an embryo, these assertions are not entirely accurate from a technical standpoint. The utility of the test is constrained by its cost and the notable incidence of false negatives [53]. It is difficult to draw a clear conclusion as to whether SJW affects the sperm-oocyte interaction and fertilization process.

Data from a review study in 2000 [54] on SJW reported that Hypericum has been linked to sperm motility impairment and probable genetic harm. High quantities of hypericum extract reduce sperm motility and viability and appear to cause genetic material mutations. However, the clinical implications of these findings remain unclear. Somatic cell mutagenesis has not been observed in embryonic animals [54]. In 2008, an animal study on the somatic and germ cells of Swiss Albino Mice reported a high dose (1520 mg/kg/day) of SJW supplementation (A Complex Mixture of SWJ, Rosemary, and Spirulina). Under experimental settings, the administered dose is typically six times higher than the calculated value [32]. This variation is attributed to the comparatively higher metabolic rate of mice than that of humans [55]. The result found SJW supplementation induced a substantial increase in the frequency of banana-shaped, swollen achrosomes and triangle-head-type sperm abnormalities (*p* < 0.05). At this dose, the percentage of total spermatozoa abnormalities was substantially higher than that in the control group (*p* < 0.05). These alterations could be attributed to the combined action of SJW terpenes, tannins, quercetin, and flavonoids [32]. Further investigations, such as sequencing of the mutated gene, are warranted to validate the nature of harm inflicted on sperm cells by SJW use.

There are no publications exploring the efficacy of SJW antioxidant properties in reproductive function. A chemical experimental study investigating nine different types of Hypericum species yielded the highest level of Deoxyribonuclease I (DNase I) inhibition potency among SJW (*Hypericum perforatum*) extract compared to other types, as this study showed the SJW extract is the most potent DNase I inhibitor, it could have a protective role towards the Deoxyribonucleic acid (DNA) and considered potentially prevent or decelerate age-related DNA fragmentation, thereby safeguarding the elderly from the onset of numerous diseases triggered by age-related apoptosis [34]. Given that oxidative stress is a key contributor to male infertility, antioxidants play a significant role in the prevention and treatment of this condition [56]. Hypericum species display noteworthy antioxidant capabilities [57,58]. In addition to their DNase I inhibitory attributes, these properties position them as substantial dietary supplements for preventing and managing the conditions resulting from both oxidative and apoptotic DNA fragmentation. Notably, it is worth mentioning that among the examined Hypericum species, *H. perforatum* demonstrated the highest antioxidant activity [36] as well as the most potent DNase I inhibition activity [34]. Because apoptotic DNA fragmentation in sperm cells could potentially play a role in the progression of male infertility, the inhibition of DNase I, a major endonuclease implicated in DNA fragmentation during apoptosis, presents an additional potential mechanism for averting male infertility [59].

One study found that SJW has pro-oxidant activity at high doses and antioxidant capabilities at low doses [60]. However, the levels of polyphenols and antioxidant capacity in *H. perforatum* L. extract standardized with HF and hypericin might differ based on the timing of harvest [61], and differences in geographic origin and biological source can result in significant alterations in the chemical composition and, subsequently, in the in vitro and in vivo biological activities of SJW preparations [62]. This should be attributed to the future direction of clinical studies related to anti-oxidant efficacy in male reproductive health.

Collectively, the available information regarding the effect of SJW on sperm motility is sparse and contradictory. Further investigation is required to gain a clearer understanding of SJW’s potential influence on male reproductive capabilities and to ascertain any possible advantages and drawbacks. Prior to considering supplements or medications, it is crucial to engage in discussions with healthcare professionals regarding fertility or reproductive health concerns.

### 4.3. Adverse Events of SJW and Precautions

One study compared SJW with conventional antidepressants, including SSRIs and TCAs. The common adverse effects of SJW were comparable to those of conventional antidepressants, with the added benefits of low cost and minimal withdrawal symptom rates [63]. Although categorized solely as a dietary supplement by the Food and Drug Administration (FDA) and not classified as a drug [63], it is important to mention that while SJW has not obtained FDA approval owing to safety concerns, FDA-approved drugs with similar actions and side-effect profiles as SJW have been available for a long time [64]. Similar to any herbal remedy, SJW toxicity studies have found that it causes erythema, edema, alopecia, weight loss, increased sensitivity to sunlight, and blood chemistry alterations, in addition to skin reddening, itching, dizziness, constipation, exhaustion, anxiety, and tiredness in both humans and animals [65,66].

In terms of side effect occurrence, in an RCT comparison of 240 subjects, fluoxetine: 114 (48%), hypericum: 126 (52%) among patients with mild to moderate depression showing hypericum had higher superiority in the total incidence, the number of patients experiencing side effects, and specific types of reported side effects [67]. Recent data from a systematic review in 2016 reported that more individuals using antidepressants reported adverse events in the included RCTs comparing SJW to standard antidepressant medicines (OR: 0.67; CI: 0.56, 0.81; 11 RCTs). SJW was linked with fewer adverse events in the gastrointestinal (OR: 0.43; CI: 0.34, 0.55; 15 RCTs) and neurologic organ systems (OR: 0.29; CI: 0.24, 0.36; 15 RCTs). Adverse events, including psychiatric or SF, also decrease in patients with SJW; however, only a few studies have reported these symptoms. Serious adverse events were not statistically different between treatment regimens (OR: 0.62; CI: 0.05, 5.46; 4 RCTs), although this conclusion was also based on a small number of studies [68]. SJW should not be combined with warfarin, cyclosporine, theophylline, digoxin, HIV protease inhibitors, anticonvulsants, SSRIs, triptans, or oral contraceptives (OCPs) [2]. Concerns have been raised about the possible interactions between SJW and previously mentioned medications. Patients taking these medications are advised to quit SJW, usually after consulting a medical expert, because dose modification of conventional treatment may be required [2]. SJW should not be initiated by patients already using any of the aforementioned medications, and patients taking other medications should be encouraged to consult a doctor before using SJW [2]. For example, SJW increases the activity of certain liver enzymes responsible for metabolizing warfarin, leading to decreased levels of warfarin in the blood and a reduced anticoagulant effect. This increased the risk of blood clots and other complications [69]. SJW has been observed to stimulate Cytochromes P450 enzymes, leading to an elevated clearance of warfarin and a reduction in the International Normalized Ratio (INR) [70], contrary to what the latter study [69] suggests about a potential INR increase. Although there have been documented decreases of 20% in the Area Under the Curve (AUC) in single-dose warfarin studies, it is advisable to maintain vigilant monitoring of INR levels. Nevertheless, it is probable that this interaction has only minor clinical significance [70].

SJW can interact with cyclosporine by increasing the activity of certain liver enzymes responsible for cyclosporine metabolism. This can lead to decreased levels of cyclosporine in the blood and a reduced immunosuppressive effect. This can increase the risk of rejection after organ transplantation or cause autoimmune disease flares [71]. SWJ interacts with theophylline by increasing the activity of liver enzymes responsible for theophylline metabolism. This can lead to decreased levels of theophylline in the blood, reducing its therapeutic effects. This can increase the risk of respiratory symptoms and exacerbation in patients with asthma or Chronic obstructive pulmonary disease (COPD) [72]. The interactions between SJW and digoxin have been supported by pharmacological studies in healthy volunteers. These investigations also showed that SJW might induce some cytochrome P450 (CYP) drug-metabolizing enzymes in the liver [73]. Furthermore, after co-administration with SWJ, the plasma concentrations of simvastatin and lovastatin decreased [74]. Finally, SJWs interfere with OCPs, reducing their effectiveness and increasing the risk of unintended pregnancy [2].

In the United States, similar to other herbal remedies, SJW is categorized as a dietary supplement by the FDA [63,75]. Consequently, it avoids the rigorous safety and efficacy assessments required for conventional pharmaceutical drugs [76]. The FDA mandates that all herbal remedies should incorporate a disclaimer, informing consumers that the therapeutic claims of the medication have not been evaluated by the agency [75].

### 4.4. Limitation

Narrative reviews are “evidence roundups” of individual healthcare topics that do not always adhere to systematic evidence-based criteria. Although narrative reviews have a greater breadth than systematic reviews, they have been criticized for their lack of coherence and rigor. Surprisingly, the available literature on SJW is limited to low-quality studies with small sample sizes and mostly animal experimental studies, supporting the need for more high-level evidence regarding the use of SJW in male sexual and reproductive health.

## 5. Conclusions

Based on our review (Table 1), the available data are related to the impact of STW on human and animal SRH. However, research on the effects of SJW on human SRH is limited. Although experimental studies have indicated the potential for SD and reproductive issues, applying these findings to humans is challenging. Drawing conclusions regarding the safety and efficacy of SJW in human subjects is difficult because of these limitations. Nevertheless, further research is required to validate these findings and ascertain the possible side effects and interactions of SJW.

Our review indicated that the safety and effectiveness of SJW in treating human SRH remain uncertain. More extensive research involving well-designed studies with larger sample sizes, extended exposure periods, and specific dosage regimens may be necessary to establish the dose-dependent effects of SJW on human SRH. It is recommended that a healthcare professional be consulted for informed decisions before considering herbal remedies or supplements.

## Figures and Tables

**Figure 1 biomedicines-11-02800-f001:**
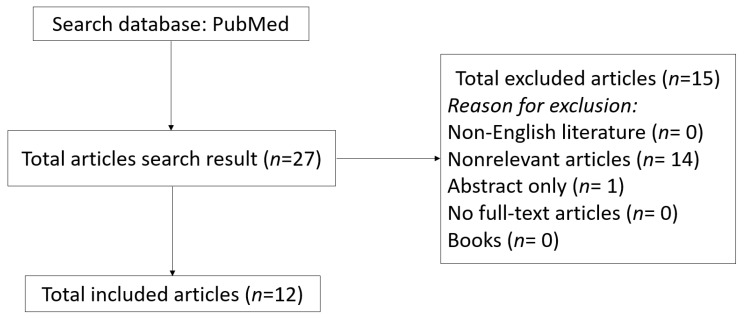
Flow diagram representing the study articles selection.

**Table 1 biomedicines-11-02800-t001:** Effect of St. John’s Wort (SJW) on Sexual and Reproductive Health.

Study/Year/Subjects	Study Design/Sample Size/Extract of SJW/Dosage	Main Outcome Related to Sexual and Reproductive Health
Ondrizek, 1999, Human [24]	In vitro study (Fresh sperm specimen), 1 (single fertile donor), NR, 0.06–0.6 mg/mL.	- SJW has a spermicidal impact and is mutagenic on human sperm cells. - SJW suppresses sperm motility. - In a laboratory setting, the presence of a small amount (0.06 mg/mL) of SJW started to hinder sperm motility after 24 h. Conversely, a larger concentration (0.6 mg/mL) of SJW had an immediate inhibitory impact on sperm motility starting from just 1 h onwards.
Ondrizek, 1999, Haman and Rats [25]	In vivo study (Fresh sperm specimen) and (hamster oocyte), 1 (single fertile donor), 12–15 hamster oocytes per group, NR, 0.06–0.6 mg/mL.	- High concentrations of SJW (0.6 mg/mL) had adverse effects on oocytes. - Pretreatment of oocytes with 0.6 mg/mL of SJW resulted in zero penetration and had inhibitory effects on the fertilization process. - Exposure of sperm to SJW resulted in DNA denaturation. - Sperm exposed to 0.6 mg/mL of SJW showed mutation of the BRCA1 exon 11 gene.
Assalian, 2000, Human [26]	Case report, 1, NR, 0.9 mg	- Erectile dysfunction - Delayed ejaculation - Orgasmic delay
Bhopal, 2001, Human [27]	Case report, 1, NR, NR	- Low libido
Lee, 2003, Human [28]	Prospective, observational, cohort study (breastfeeding women), 33, NR, NR	- No statistically significant differences were found in maternal or infant demographics or maternal adverse events. - Infant adverse events were reported in two cases of colic, two cases of drowsiness, and one case of lethargy. - No significant difference was observed in the frequency of maternal reports of decreased milk production among the groups, nor was a difference found in infant weight over the first year of life.
Gregoretti, 2004 Rats [29]	Experimental study (pregnant and breastfeeding rats), NR, SJW extracted in methanol solution and containing 0.3% total hypericin, 100–1000 mg/kg per day	- Liver and kidney damage - Chronic SJW use during pregnancy or breastfeeding causes histological changes in the liver and kidneys of rats. The impact is dose-dependent and visible with a dosage comparable to that used for depression treatment.
Capasso, 2005, Haman and Rats [30]	Experimental study, NA, Dry hydromethanolic extract from *H. perforatum* flowering top, 0.3% hypericin, 1–300 µg/ml	- Inhibition of rat and human vas deferens smooth muscle contractions leads to a delay in sperm emission, which explains delayed ejaculation.
Markowitz, 2005, Human [31]	Pilot study (healthy volunteers), 12 (6 males, 6 females), Flowers and leaves SJW extract (Kira^®^), 0.12%–0.3% hypericin, 300 mg	- SJW did not significantly alter most of the androgen. - The combined concentrations of the 5α-reduced steroids, AoS, and epiAoS significantly declined following treatment in all subjects. - The testosterone to DHT ratio was increased in all subjects but did not reach statistical significance; it is also consistent with the possible inhibition of 5α-reductase by SJW. - SJW causes significant induction of CYP3A4; however, short-term administration of SJW does not significantly alter the concentrations of most circulating androgens in men and women but may produce a reduction in some of the circulating 5α-reduced androgens.
Thomas, 2007, Rats [16]	Experimental study, 12, Hyperforin-rich extract, 5 to 80 mg/kg	- Hyperforin can delay the ejaculatory reflex
Aleisa, 2008, Mice [32]	Experimental study (Male Swiss albino mice) NR SJW supplements (Complex Mixture of SWJ, Rosemary, and Spirulina) 380, 760, and 1520 mg/kg/day	- It led to anomalies in the chromosomes within the testes and triggered abnormalities in spermatozoa such as banana-shaped, swollen achrosome and triangular head and induced chromosomal aberrations (aneuploids, polyploids, and total aberrations) in the testes.
Vieira,2013, Rats [33]	Experimental study, 9, SJW (Herbarium, Brazil), 100 mg/kg	- Maternal exposure to SJW did not interfere with reproductive parameters in adult male rats.
Kolarevic, 2019, Bovine pancreatic DNase and Rat liver homogenate [34]	Experimental study, NR, (*Hypericum perforatum*, Serbia), NR	- The most potent DNase I inhibition was obtained by water extract of *H. perforatum* and exhibited the most potent antioxidant activity. - As a DNase I inhibitor, *H. perforatum* could potentially hinder or delay age-related DNA fragmentation, consequently offering protection to the elderly against the onset of numerous diseases triggered by age-related apoptosis as well as in the prevention of male infertility.

SJW; St. John’s wort, NA; not applicable; NR; not reported, *Hypericum perforatum*; *H. perforatum*.

## Data Availability

Not applicable.

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
