# Peer review of "Effect of St. John’s Wort (Hypericum perforatum L.) on Male Sexual and Reproductive Health: A Narrative Review"

_biomedicines, 2023, doi:10.3390/biomedicines11102800_

Round 1
Reviewer 1 Report
Comments to the manuscript biomedicines-2641453 "Effect of St John’s Wort (Hypericum perforatum) on Male Sexual and Reproductive Health: A Narrative Review"
Authors propose a review of the scientific literature on the effects of the John's Wort on male sexual and reproductive health. The oldest publlication is from 1999 and only 12 articles were considered in the review. Available findings are well presented and clearly discussed. In spite of the limited results, the study is of potential interest for the field of research and may be suitable for publication after sono minor changes.
First of all, the Table 1 should be redrawn in order to avoid a so wide space without text. Probably this is possible reducing the number of columns by association of some of them and their content: e.g. first column (Study/Year/Subjects); second column (Study desing/sample size/extract/dosage), third column (outcome description).
Page 3, line 124: please change "full-text" with "no full-text"
Minor editing of English language required.
Author Response
The manuscript is interesting and summarizes the findings on Hypericum perforatum in relation to sexual and reproductive health. The introduction provides the relevant background, and the methodology used for paper selection seems to be appropriate. However, the manuscript needs refinement before consideration for publication. Especially, the Table requires more work. Detailed suggestions are listed below.
Response: we would like to thank the reviewer for his valuable feedback and comments.
- The name of the plant should be consistently written in italics. Furthermore, after its first mention in the text, the abbreviation 'H. perforatum' should be used.
Response: done
- Repetition should be avoid: see in line 51 “herbaceous perennial plant” and in line 53 “a perennial herbaceous plant”.
Response: repetition was removed.
- Line 65: “However, it is mostly used to treat depression,” – this statement is unnecessary in this place and disturb the fluency of the text
Response: this statement was removed.
- Line 67: “natural products” replace by “natural compounds” or “components”
Response: replaced with “natural compounds”
- Line 156: “The primary outcomes of the reviewed studies (Table 1).” – lack of verb.
Response: rephrase to better understanding (The reviewed studies primarily yielded outcomes in Table 1.
- Lines 152-171 – add references to support the statements
Response: we added at the end of this paragraph citation for the results
- Line 166: “that SJW, particularly Hypericum perforatum,”- The authors use 'Hypericum perforatum' and 'SJW' as synonyms, so I do not understand this expression.
Response: the line was removed and replaced with “the H. perforatum”
8) Form of table needs refinement. Maybe You could combined Subjects and sample size to reduce the number of column. Reduce the number of words in last column. Some of them are unnecessary. For example “High concentrations of SJW (0.6 mg/mL)” – “0.6 mg/mL” is enough. Explain the abbreviation NR/NA
Response: done. Also based on reviewer 1 he ask the same.
- Usually future directions are included to Conclusion section
Response: done.

Reviewer 2 Report
The manuscript is interesting and summarizes the findings on Hypericum perforatum in relation to sexual and reproductive health. The introduction provides the relevant background, and the methodology used for paper selection seems to be appropriate. However, the manuscript needs refinement before consideration for publication. Especially, the Table requires more work. Detailed suggestions are listed below.
1) The name of the plant should be consistently written in italics. Furthermore, after its first mention in the text, the abbreviation 'H. perforatum' should be used.
2) Repetition should be avoid: see in line 51 “herbaceous perennial plant” and in line 53 “a perennial herbaceous plant”.
3) Line 65: “However, it is mostly used to treat depression,” – this statement is unnecessary in this place and disturb the fluency of the text
4) Line 67: “natural products” replace by “natural compounds” or “components”
5) Line 156: “The primary outcomes of the reviewed studies (Table 1).” – lack of verb.
6) Lines 152-171 – add references to support the statements
7) Line 166: “that SJW, particularly Hypericum perforatum,”- The authors use 'Hypericum perforatum' and 'SJW' as synonyms, so I do not understand this expression.
8) Form of table needs refinement. Maybe You could combined Subjects and sample size to reduce the number of column. Reduce the number of words in last column. Some of them are unnecessary. For example “High concentrations of SJW (0.6 mg/mL)” – “0.6 mg/mL” is enough. Explain the abbreviation NR/NA
9) Usually future directions are included to Conclusion section
Author Response
Reviewer 2
The manuscript is interesting and summarizes the findings on Hypericum perforatum in relation to sexual and reproductive health. The introduction provides the relevant background, and the methodology used for paper selection seems to be appropriate. However, the manuscript needs refinement before consideration for publication. Especially, the Table requires more work. Detailed suggestions are listed below.
Response: we would like to thank the reviewer for his valuable feedback and comments.
- The name of the plant should be consistently written in italics. Furthermore, after its first mention in the text, the abbreviation 'H. perforatum' should be used.
Response: done
- Repetition should be avoid: see in line 51 “herbaceous perennial plant” and in line 53 “a perennial herbaceous plant”.
Response: repetition was removed.
- Line 65: “However, it is mostly used to treat depression,” – this statement is unnecessary in this place and disturb the fluency of the text
Response: this statement was removed.
- Line 67: “natural products” replace by “natural compounds” or “components”
Response: replaced with “natural compounds”
- Line 156: “The primary outcomes of the reviewed studies (Table 1).” – lack of verb.
Response: rephrase to better understanding (The reviewed studies primarily yielded outcomes in Table 1.
- Lines 152-171 – add references to support the statements
Response: we added at the end of this paragraph citation for the results
- Line 166: “that SJW, particularly Hypericum perforatum,”- The authors use 'Hypericum perforatum' and 'SJW' as synonyms, so I do not understand this expression.
Response: the line was removed and replaced with “the H. perforatum”
8) Form of table needs refinement. Maybe You could combined Subjects and sample size to reduce the number of column. Reduce the number of words in last column. Some of them are unnecessary. For example “High concentrations of SJW (0.6 mg/mL)” – “0.6 mg/mL” is enough. Explain the abbreviation NR/NA
Response: done. Also based on reviewer 1 he ask the same.
- Usually future directions are included to Conclusion section
Response: done.

Reviewer 3 Report
The authors of the manuscript assessed the effect of St. John's wort on sexual function and sexual health in men. The work will be reviewed, and as such it is carried out properly and properly discussed.
I have a few general comments that I would like to include (preferably in the introduction):
- St. John's wort used in patients with depressive illness may be beneficial, but please specify and clearly emphasize that St. John's wort treatment is only for some patients. assessment by a psychiatrist before inclusion is necessary.
- many interactions with various drugs - for example, not so long ago it was believed that the DOAC group was safe in this respect, which is not true. please pay attention to the clinical significance of drug–food interactions of direct oral anticoagulants. What are the consequences and what measures should be taken (e.g. TDM!)
- when assessing sexual function, especially in people with comorbidities, it is sometimes difficult to distinguish which comes first: the egg or the chicken... Please emphasize this in the introduction
Author Response
Reviewer 3
The authors of the manuscript assessed the effect of St. John's wort on sexual function and sexual health in men. The work will be reviewed, and as such it is carried out properly and properly discussed.
I have a few general comments that I would like to include (preferably in the introduction):
Response: we would like to thank the reviewer for his valuable feedback and comments.
- St. John's wort used in patients with depressive illness may be beneficial, but please specify and clearly emphasize that St. John's wort treatment is only for some patients. assessment by a psychiatrist before inclusion is necessary.
Response: we added a statement for this regard in discussion section line 251-253
- many interactions with various drugs - for example, not so long ago it was believed that the DOAC group was safe in this respect, which is not true. please pay attention to the clinical significance of drug–food interactions of direct oral anticoagulants. What are the consequences and what measures should be taken (e.g. TDM!)
Response: we added a statement (lines 477-483) for monitoring anticoagulants such as warfarin interaction with SJW, and we mentioned the way of monitoring based on one new reference.
- when assessing sexual function, especially in people with comorbidities, it is sometimes difficult to distinguish which comes first: the egg or the chicken... Please emphasize this in the introduction
Response: we added a statement for regard between lines 114 to 119.

Round 2
Reviewer 2 Report
I have no additional suggestions
Reviewer 3 Report
the authors have significantly modified the manuscript, in my opinion it may be considered for publication